# Dexamethasone-Loaded Hydrogels Improve Motor and Cognitive Functions in a Rat Mild Traumatic Brain Injury Model

**DOI:** 10.3390/ijms231911153

**Published:** 2022-09-22

**Authors:** Christian Macks, Daun Jeong, Sooneon Bae, Ken Webb, Jeoung Soo Lee

**Affiliations:** 1Drug Design, Development, and Delivery (4D) Laboratory, Department of Bioengineering, Clemson University, Clemson, SC 29615, USA; 2MicroEnvironmental Engineering Laboratory, Department of Bioengineering, Clemson University, Clemson, SC 29615, USA

**Keywords:** hydrogel, dexamethasone, traumatic brain injury, neuroinflammation, apoptosis, motor function, cognitive function

## Abstract

Functional recovery following traumatic brain injury (TBI) is limited due to progressive neuronal damage resulting from secondary injury-associated neuroinflammation. Steroidal anti-inflammatory drugs, such as dexamethasone (DX), can reduce neuroinflammation by activated microglia and infiltrated macrophages. In our previous work, we developed hydrolytically degradable poly(ethylene) glycol-bis-(acryloyloxy acetate) (PEG-bis-AA) hydrogels with dexamethasone (DX)-conjugated hyaluronic acid (HA-DXM) and demonstrated that dexamethasone-loaded hydrogels (PEG-bis-AA/HA-DXM) can reduce neuroinflammation, apoptosis, and lesion volume and improve neuronal cell survival and motor function recovery at seven days post-injury (DPI) in a rat mild-TBI model. In this study, we investigate the effects of the local application of PEG-bis-AA/HA-DXM hydrogels on motor function recovery at 7 DPI and cognitive functional recovery as well as secondary injury at 14 DPI in a rat mild-CCI TBI model. We observed that PEG-bis-AA/HA-DXM-treated animals exhibit significantly improved motor functions by the rotarod test and cognitive functions by the Morris water maze test compared to untreated TBI animals. We also observed that PEG-bis-AA/HA-DXM hydrogels reduce the inflammatory response, apoptosis, and lesion volume compared to untreated animals at 14 DPI. Therefore, PEG-bis-AA/HA-DXM hydrogels can be promising a therapeutic intervention for TBI treatment.

## 1. Introduction

Traumatic brain injuries remain a serious health concern worldwide, with recent estimates reporting that 10–60 million new cases occur annually [1,2]. These injuries often lead to long-term disabilities and approximately 2.5 million people in the United States are living with TBI-related disabilities [3]. The initial trauma results in a primary injury that includes cerebral contusion, axonal shearing, and damage to vasculature [4,5,6,7]. This initial damage provokes the release of excitatory neurotransmitters, calcium influx into neurons, and the initiation of a secondary injury phase characterized by progressive neuroinflammatory responses that lead to delayed cell death and tissue damage [8,9,10]. The primary injury damage is rapid and is often considered as preventable but not treatable. Therefore, therapeutic interventions for improving functional recovery focus on limiting delayed cellular apoptosis and progressive neuroinflammation in the secondary injury phase [10,11].

Pro-inflammatory cytokines produced by activated microglia and macrophages, such as IL-1β and TNFα, drive inflammatory activation and promote cellular apoptosis. Suppressing excessive inflammation following injury remains a key target for improving functional recovery [11]. Glucocorticoids (GCs) have been used to control edemas and inflammation in clinical TBI patients [12,13]. Synthetic glucocorticoids (i.e., methylprednisolone, dexamethasone, etc.) exert their effects through the activation and nuclear translocation of GC receptors [14]. Dexamethasone (DX) has demonstrated a significant reduction in early pro-inflammatory cytokine expression when used in high-dose systemic administration, and the use of DX for the treatment of head trauma spans 40 years [12]. However, the long-term, high-dose administration of DX is also associated with side-effects, such as diabetes, glaucoma, and osteoporosis [12]. Additionally, the CRASH international randomized double-blind clinical trial reported increased mortality [15] from high-dose synthetic GC administration in the treatment of moderate to severe TBI. Other studies have observed no significant benefits of high-dose GC treatment [16,17,18]. These have considerably contra-indicated the use of DX in recent clinical guidelines [13,19]. However, several studies reported that low-dose DX can reduce secondary injury and improve functional recovery [20,21,22].

In our previous papers, we synthesized semi-interpenetrating network (semi-IPN) hydrogels, PEG-bis-AA/HA-DXM, composed of poly(ethylene glycol)-bis-(acryloyloxy acetate) (PEG-bis-AA) and DX-conjugated hyaluronic acid (HA) via a monosuccinate linker (HA-DXM) [23]. We demonstrated that the local application of PEG-bis-AA/HA-DXM hydrogels (~3 µg DX/gel) to the injured brain can reduce neuroinflammation and apoptosis post-TBI while improving neuronal cell survival and motor function in a rat mild-controlled-cortical-impact (CCI) model of TBI at seven days post-injury (DPI) [24]. Although our study was able to demonstrate therapeutic efficacy, it was at a very critical seven DPI time point, and the potential benefits to cognitive-function recovery were not analyzed. Here, we extend our investigation of the therapeutic efficacy of PEG-bis-AA/HA-DXM to a longer time period (14 DPI) and add a cognitive functional assessment. Motor function is evaluated by a rotarod test at 7 DPI and cognitive function by the Morris water maze (MWM) test at 14 DPI. Upon completion of the cognitive assessment, tissue samples were collected for 1) the histological analysis of the effect of hydrogels on neuroinflammation, apoptosis, lesion volume, and neuronal survival, and 2) the analysis of inflammatory gene mRNA expression.

## 2. Results

### 2.1. PEG-bis-AA/HA-DXM Hydrogels Improve Motor Function after TBI

The effect of PEG-bis-AA/HA-DXM treatment on motor function was evaluated by an accelerating rotarod test (4 rpm to 40 rpm over 90 s) at 1, 3, and 6 DPI (Figure 1). The latency time on the rotating rod was significantly different in both the untreated TBI and PEG-bis-AA/HA-DXM animal treatment groups compared to the sham group at all time points. However, we observed that the latency time on the rotating rod was longer in the PEG-bis-AA/HA-DXM-treatment group at all time points and significantly different from the untreated TBI group at 6 DPI.

### 2.2. PEG-bis-AA/HA-DXM Hydrogels Improve Cognitive Function by Morris Water Maze Test

The effect of PEG-bis-AA/HA-DXM treatment on cognitive function was evaluated by a Morris water maze test for spatial learning starting at 8 DPI (Figure 2). Testing consisted of four trials a day for five consecutive days. The time required for the rat to find the hidden platform and stay on the platform for 1 s was recorded as latency time. The latency time in the PEG-bis-AA/HA-DXM-treatment group was shorter than that in the untreated TBI group at all time points and significantly different at 12 DPI (Figure 2). We also observed that the latency time for the PEG-bis-AA/HA-DXM-treatment group was not significantly different from the sham group at 10 and 12 DPI (Figure 2). This result indicates that PEG-bis-AA/HA-DXM treatment improved spatial memory function following TBI.

### 2.3. Effect of PEG-bis-AA/HA-DXM Hydrogels on Lesion Volume

At 14DPI, following the completion of behavioral studies, animals (*n* = 5 rats per group) were sacrificed by cardiac perfusion under deep anesthesia using isoflurane gas. The brains were collected, fixed, and sectioned for histological analysis. Lesion volumes were measured following Nissl body staining using cresyl violet. For staining, 10 sections with even 0.25 µm spacing were selected and stained, as described in the Methods Section. Figure 3A presents the representative images of stained Nissl bodies in the brain sections. The lesion volume was calculated by Cavalieri approximation. We observed that the average lesion volume in the PEG-bis-AA/HA-DXM (11.9 ± 1.84 mm^3^)-treatment group was less than that in the untreated TBI group (13.3 ± 2.36 mm^3^), even though it was not significantly different (Figure 3B).

### 2.4. Effect of PEG-bis-AA/HA-DXM Hydrogels on Neuroinflammatory Response

At 14 DPI, following the behavior study, one set of animals (*n* = 5) were sacrificed for histological analysis and the other set of animals (*n* = 3) were sacrificed for the analysis of gene expression by RT-PCR.

#### 2.4.1. Expression of Inflammatory Cytokine Genes

To investigate the effects of PEG-bis-AA/HA-DXM treatment on neuroinflammation at 14 DPI, the mRNA expression levels of IL1β, IL10, TGFβ1, TNFα, and Ifn*γ* were measured by RT-PCR (Figure 4A). We observed that the relative expression levels of the pro-inflammatory cytokines IL1β, TGFβ1, TNFα, and Ifn*γ* in the PEG-bis-AA/HA-DXM-treatment group were significantly lower compared to those in the untreated TBI group. The expressions of IL1β, TGFβ1, TNFα, and Ifn*γ* were not significantly different in the PEG-bis-AA/HA-DXM-treatment group compared to the sham control group (Figure 4A). This result indicates that DX release from PEG-bis-AA/HA-DXM hydrogels can provide a long-term suppression of inflammatory signaling following TBI.

#### 2.4.2. Histological Analysis of Activated Microglia/Infiltrated Macrophages

The effect of PEG-bis-AA/HA-DXM treatment on the inflammatory response was analyzed by the detection of CD68 clone-ED1-positive (ED1+)-activated microglia/infiltrated macrophages at 14 DPI (Figure 4B,C). The percentage of ED1+ cells in the PEG-bis-AA/HA-DXM-treated group was significantly lower compared to that in the untreated “TBI group at 14 DPI (Figure 4B). Figure 4C presents representative images of ED1+ cells in the stained ipsilateral brain section.

### 2.5. Effect of PEG-bis-AA/HA-DXM on Apoptosis in the Ipsilateral Cortex

#### 2.5.1. Gene Expression of BAX and Bcl-2

To investigate the effect of PEG-bis-AA/HA-DXM treatment on apoptosis, we measured the mRNA expression levels of the pro-apoptotic protein BAX and the anti-apoptotic protein Bcl-2 by RT-PCR at 14 DPI (*n* = 3 rats/group). The ratio of BAX to Bcl-2 expression levels was significantly increased in the untreated TBI compared to the sham control group, while the BAX-to-Bcl-2 ratio in the PEG-bis-AA/HA-DXM-treatment group was not significantly different compared to the sham group (Figure 5A). We observed that PEG-bis-AA/HA-DXM treatment significantly reduced the BAX-to-Bcl2 ratio compared to the untreated TBI group (Figure 5A). This result indicates that the sustained release of X from PEG-bis-AA/HA-DXM hydrogels suppresses apoptotic signaling following TBI.

#### 2.5.2. Apoptotic Response by TUNEL Assay

The effect of PEG-bis-AA/HA-DXM treatment on apoptotic response was also evaluated by TUNEL staining at 14 DPI. The percentage of TUNEL positive-cell nuclei was significantly reduced in the PEG-bis-AA/HA-DXM-treated group compared to the untreated TBI group (Figure 5B). Figure 5C presents representative images of TUNEL-stained brain sections, and TUNEL ^+^ cells can be observed at the lesion border.

### 2.6. Effect of PEG-bis-AA/HA-DXM Hydrogels on Neuronal Cell Survival in the Ipsilateral Cortex

The effect of PEG-bis-AA/HA-DXM treatment on neuronal cell survival was evaluated by immunohistochemistry (IHC) for neuron-specific nuclear proteins (NeuNs). The number of NeuN-positive (NeuN+) cell nuclei was significantly reduced in both the PEG-bis-AA/HA-DXM-treated and untreated TBI groups compared to the sham control group (Figure 6A). There was no significant difference in the number of NeuN+ cells between the PEG-bis-AA/HA-DXM-treated and untreated TBI groups. Figure 6B presents the representative images of NeuN-stained brain sections.

## 3. Discussion

The initial mechanical trauma and primary injury that result from TBI subsequently initiate a chronic secondary injury phase that is characterized by widespread neuroinflammation, excitotoxicity, and oxidative stress. Progressive neuroinflammation contributes to secondary neuronal cell death after TBI, and resident microglia become activated by release of stress-related chemokines and apoptotic elements following injury [25]. Additionally, vascular damage by the primary injury allows the entry of peripheral immune cells, including neutrophils, monocyte-derived macrophages, and lymphocytes [26,27]. Activated microglia, and to a greater degree peripheral macrophages [28], secrete proinflammatory cytokines, such as IL1β, IL6, IL12, and TNFα, which increase inflammation and apoptotic activity, thereby leading to progressive and widespread damage to neural circuitry [25].

The treatment of severe TBI with synthetic glucocorticoids, including dexamethasone (DX), has experienced clinical use for attenuating neuroinflammation and edema [12,29]; however, the high-dose systemic administration of synthetic glucocorticoids is associated with serious side-effects in rat models of brain injury [30,31], and is not recommended for improving the outcome of severe TBI in humans based on multicenter clinical evaluation data [15,19]. In contrast, the low-dose systemic administration of DX is associated with decreased neuroinflammation and improved outcomes in rat brain-injury models [20,21,22]. In our previous study, we observed similar effects of low doses of local DX delivery from hydrolytically degradable PEG-bis-AA/HA-DXM hydrogels [24]. We demonstrated that locally delivered low-dose DX (~3 µg DX/gel) can reduce lesion volume, neuroinflammatory response, and cellular apoptosis, and improve motor function recovery at 7 DPI following mild TBI in a rat CCI model.

In this study, the therapeutic efficacy of PEG-bis-AA/HA-DXM on motor function up to 7 DPI and cognitive function at 14 DPI was assessed. With regard to motor function, we observed that the latency time to fall was longer in the PEG-bis-AA/HA-DXM-treatment group relative to the untreated TBI group up to 6 DPI (Figure 1). This was consistent with the trend we observed in our previous study using a beam-walk test [24]. For cognitive function, we observed that the latency time in the PEG-bis-AA/HA-DXM-treatment group was shorter than that in the untreated TBI group at all time points, and significantly different at 12 DPI (Figure 2). In contrast, several other studies using high-dose systemic doses of DX showed that treatment with high-dose DX led to worsened cognitive function and longer searching times for the hidden platform in the Morris water maze test [30,31,32]. Zhang et al. reported that sustained treatment with high-dose intraperitoneal injections of DX (5 mg/kg) once a day for 3 days led to longer search times in the water maze test and worsened neurological scores from 8 to 12 DPI [30]. A separate study using an intraperitoneal injection of methylprednisolone (5 mg/kg or 30 mg/kg) for 4 days observed a similarly negative outcome in spatial learning and cognitive function related to decreased hippocampal synaptic plasticity in a fluid percussion injury TBI model in rats [33]. Therefore, we believe that the sustained local delivery of low-dose DX using hydrogel has potential as a therapeutic strategy to treat TBI.

Histological analysis showed that the lesion volume in the PEG-bis-AA/HA-DXM-treatment group decreased compared to that in the untreated TBI group at 14 DPI, but the difference was not statistically significant (Figure 3). This was an unexpected result considering that we observed that the lesion volume was significantly reduced in the PEG-bis-AA/HA-DXM-treated group compared to the untreated TBI group at 7 DPI [24]. Agoston et al. reported that the edema after injury in rodent TBI reached a peak at 2 DPI and then declined between 7 to 14 DPI [34]. We assumed that the effects of PEG-bis-AA/HA-DXM on lesion volume may appear more accentuated at 7 DPI when closer to a peak edema rather than at 14 DPI.

The upregulation of pro-inflammatory cytokines *IL1β, TNFα*, and *IFN**γ* following TBI is well characterized [26,35] and was observed in our untreated TBI group relative to the sham control. Proinflammatory cytokine expression levels were significantly reduced in the animal groups treated with PEG-bis-AA/HA-DXM hydrogels. In addition, the hydrogel treatment significantly reduced the expression of TGFβ1 compared to the untreated TBI group. The role of *TGFβ1* following brain injury has both positive [35,36,37,38] and negative [39,40] aspects and its associations with alternatively activated, anti-inflammatory macrophage/microglial phenotypes [25,36,37,41] indicates its importance in maintaining a “healthy” immune response. Therefore, it is important to note that while *TGFβ1* expression was significantly reduced by PEG-bis-AA/HA-DXM treatment, the expression level was not significantly lower than the uninjured sham control (Figure 4A).

The resolution of the inflammatory response is typically characterized by a decrease in pro-inflammatory cytokine expression in tandem with an increase in the expression of the anti-inflammatory cytokines, and the increased expression of the anti-inflammatory cytokine *IL10* is associated with the alternative activation of microglia/macrophages [25,42]. We observed a significant increase in *IL10* expression following TBI (Figure 4A), which was to be expected as part of a severe injury-related inflammatory response. In the present study, *IL10* expression in the PEG-bis-AA/HA-DXM group was also significantly reduced compared to the untreated TBI group and remained only slightly elevated compared to the uninjured sham group at 14 DPI (Figure 4A). In our previous investigation conducted at 7 DPI, the expression of *IL10* remained significantly elevated compared to the uninjured sham group, while being significantly reduced compared to the untreated TBI group [24]. Several studies have reported that early *IL10* upregulation is associated with decreased levels of TNFα and encourages the resolution of the inflammatory response [25,43]. In our histological analysis, we observed significantly fewer ED1+ cells in the PEG-bis-AA/HA-DXM-treated group compared to that in the untreated TBI group (Figure 4B,C). Together with the cytokine expression data, these results could indicate that the sustained, local delivery of a low dose of DX from hydrogels can mitigate the inflammatory response.

Apoptosis is a key aspect of chronic secondary injury and is worsened by prolonged neuroinflammation. We evaluated the effect of PEG-bis-AA/HA-DXM hydrogels on apoptosis by RT-PCR analysis for gene expression and TUNEL staining for the apoptotic cells (Figure 5). The ratio of gene expression between pro-apoptotic gene *BAX* and anti-apoptotic gene *Bcl-2* is a method frequently used for determining the level of apoptotic activity in cells [44,45,46,47,48,49]. We observed that the ratio of *BAX* to *Bcl-2* expression was significantly reduced in the PEG-bis-AA/HA-DXM group compared to the untreated TBI and sham control groups (Figure 5A). These results are consistent with our previous study using PEG-bis-AA/HA-DXM hydrogels at 7 DPI [24] and with similar observations that DX administration can reduce pro-apoptotic *BAX* gene expression while supporting the expression of anti-apoptotic *Bcl2* gene expression [44,45,46,47,48,49]. We also observed that the percentage of TUNEL+ cell nuclei in the perilesional cortex was significantly reduced in the PEG-bis-AA/HA-DXM-treatment group compared to the untreated TBI group (Figure 5B).

For evaluating neuronal cell survival, we performed immunohistochemical staining using an NeuN antibody against neuronal cell nuclei. Unexpectedly, the number of NeuN+ cells in the PEG-bis-AA/HA-DXM-treatment group was not significantly higher than that in the untreated TBI group, although we observed a significant reduction in the number of apoptotic TUNEL+ nuclei (Figure 6).

The observed benefits of PEG-bis-AA/HA-DXM treatment on improving cognitive functional recovery and decreasing neuroinflammation and apoptosis at 14 DPI suggest that a sustained, low-dose delivery of DX from this hydrogel platform may be an effective treatment for TBI. We believe that the benefits of PEG-bis-AA/HA-DXM treatment on motor and cognitive functional recovery are most likely due to mitigating the inflammatory response following TBI. Table 1 summarizes our key findings on the effect of PEG-bis-AA/HA-DXM hydrogel treatment from our previous study at 7 DPI and the current study at 14 DPI. Overall, these results show a consistent and sustained therapeutic effect. One limitation of these studies was that we evaluated the therapeutic effects of immediate (<30 min), local treatment with PEG-bis-AA/HA-DXM hydrogels following TBI as a proof of concept; however, in clinical settings, the treatment is often more delayed. A study conducted by Sun et al. in a rat ischemia/reperfusion-injury model demonstrated that the therapeutic effects of systemic DX administration can be strongly influenced by the onset of treatment [50]. Sun et al. reported that the intravenous administration of DX attenuated the infarct size and neurological deficit when delivered within 30 min of middle-cerebral-artery occlusion; however, treatment with DX was not significantly different from injury alone when treatment onset was delayed by 60 min or more [50]. Although this evaluation was conducted in an ischemic-injury model and with high-dose DX (10 mg/kg), it supported the conclusion that future evaluations of PEG-bis-AA/HA-DXM treatment in TBI should investigate hydrogel application at more clinically relevant delayed post-injury time points (>1 h).

## 4. Materials and Methods

### 4.1. Materials

The 4 kDa poly(ethylene glycol) (PEG) was purchased from Fluka (Buchs, Switzerland). Sodium hyaluronate (1.5 MDa) was purchased from Lifecore Biomedical (Chaska, MN, USA). Dexamethasone (DX), chloroacetyl chloride, 1,1′-carbonyldiimidazole (CDI), sodium acrylate, triethylamine (TEA), 4-methoxyphenol, anhydrous sodium sulfate, sodium bicarbonate, and sodium chloride were purchased from Sigma-Aldrich (Milwaukee, WI, USA). Dowex 50WX8 200-400 (H) was purchased from Alfa Aeser (Ward Hill, MA, USA). Anhydrous dichloromethane, anhydrous dimethylformamide, HPLC-grade acetonitrile, chloroform, ethyl ether, and dimethyl sulfoxide (DMSO) were purchased from Fisher Chemical (Fair Lawn, NJ, USA). The 2-hydroxy-1-[4-(hydroxyethoxy) phenol]-2-methyl-1-propanone (Irgacure 2959, I-2959) was donated by BASF (Florham Park, NJ, USA).

### 4.2. Synthesis and Preparation of PEG-bis-AA/HA-DXM Hydrogels

PEG-bis-acryloyloxy acetate (PEG-bis-AA) was synthesized by a two-step process, as previously described [51]. Briefly, PEG-bis-chloroacetate was created by the activation of the terminal PEG hydroxyl groups using chloroacetyl chloride. This PEG-bis-chloroacetyl acetate product was purified and then reacted with sodium acrylate to obtain PEG-bis-AA. The structure of PEG-bis-AA was assessed by ^1^H NMR (Bruker 300 MHz, CDCl_3_, Billerica, MA, USA).

DX was conjugated to hyaluronic acid (HA-DXM) by a two-step process, as previously reported [23]. Briefly, the hydroxyl group of DX was reacted with succinic anhydride to produce dexamethasone monosuccinate (DXM). Then, the carboxyl group of DXM was activated by 1,1′-carbonyldiimidazole in the presence of triethylamine. The resulting product was mixed with HA to conjugate DXM to the hydroxyl groups of HA. The HA-DXM structure was evaluated by ^1^H NMR using a D_2_O/DMSO mixture (1:4 *v*/*v*) and an FTIR spectrometer (Thermo-Nicolet Magna-IR 550, Thermo Scientific, Waltham, MA, USA) equipped with a Thermo-SpectraTech Foundation Series Endurance Diamond ATR.

PEG-bis-AA/HA-DXM hydrogels were prepared as previously reported [23]. The PEG-bis-AA macromer was first dissolved in PBS at 15% *w*/*v*. The HA-DXM solution was prepared at 4% *w*/*v* in PBS. To produce semi-interpenetrating-network (semi-IPN) hydrogels, a gel solution was prepared by mixing PEG-bis-AA solution (final 6% *w*/*v*), HA-DXM solution (final 0.4% *w*/*v*), and I-2959 (final 0.1% *w*/*v*), and filtered using a 0.2 µm syringe filter. The 25 μL gel solution was placed between glass microscope coverslips separated by Teflon spacers (thickness: 0.75 mm) and photo-polymerized for 5 min on each side by exposure to low-intensity UV light (365 nm, 10 mW/cm^2^; Black-Ray B100-AP, Upland, CA, USA). Gel discs with a 4 mm diameter and 0.75 mm thickness were obtained.

### 4.3. Generation of a Rat Mild Traumatic Brain Injury Model and Hydrogel Treatment

Male Sprague Dawley rats (starting weight: 350–400 g) were purchased from Charles Rivers Laboratories (Wilmington, MA, USA) and housed under pathogen-free conditions at the Godley-Snell Research Center (Clemson, SC, USA). The animals were provided with a standard diet and water ad libitum. All animal handling, surgical procedures, and post-operative care were performed according to NIH guidelines for the care and use of laboratory animals and under the guidance of the Clemson University Animal Research Committee (IACUC protocol #2015-082). To generate the mild-CCI TBI model, rats were anesthetized with 2–4% isoflurane gas using a nose cone and the top of their heads were shaved. Each rat was secured in a stereotaxic frame (David Kopf Instruments, Tujunga, CA, USA) and the scalp sterilized by betadine. A midline incision was made in the scalp, the skin retracted, and the soft tissue debrided from the skull surface. Without disturbing the underlying dura, a circular craniotomy was performed over the right cortex at 3.5 mm posterior and 3.5 mm lateral of the bregma point using a dental drill equipped with a 5 mm diameter trephine bur tip. The skull cap was carefully removed and placed into sterile saline. An impact was made to the exposed dura mater above the cortex using a TBI impactor (Precision Systems and Instruments, Fairfax Station, VA, USA) device with a flat, circular impacting tip (3 mm diameter) at a speed of 3.5 m/s, depth of 2 mm, and dwell time of 250 ms. Immediately following the impact, hemostasis was achieved with GelFoam^®^ Sterile Sponge (Patterson Vet Generics, Devens, MA, USA). Following hemostasis, PEG-bis-AA/HA-DXM hydrogels (4 mm diameter and 0.75 mm thickness) were placed on the injured cortex and the skull cap was carefully placed on top of the gel before the scalp was sutured closed with black 4-0 silk suture. The skull cap was not secured with bone cement or dental acrylic to allow for tissue expansion and relieve intracranial pressure. For untreated TBI rats, the skull cap was placed on top of the dura mater after impact. For the sham surgery group, rats only received a craniotomy without impact. We did not include a PEG-bis-AA/HA hydrogel in the DX group because we did not observe significant therapeutic benefits nor any toxic side-effects of vehicle only in our previous study at 7 DPI [24]. After the motor and cognitive function test, one set of rats (*n* = 3/group) was sacrificed for cytokine gene expression analysis by RT-PCR and the other set of rats (*n* = 5/group) were sacrificed for histological analysis after a function study at 14 DPI.

### 4.4. Rotarod Test

An accelerating rotarod test was performed at 1, 3, and 6 DPI to evaluate the effect of PEG-bis-AA/HA-DXM treatment on motor-function recovery. The rotarod apparatus (Panlab, Harvard Apparatus, Cambridge, MA, USA) consisted of a textured rotating rod set to accelerate from 4 rpm to 40 rpm over 90 sec. Rats were acclimated to rotarod testing by training 3 days prior to injury. Rats were placed on the rod starting at a constant speed of 4 rpm and acceleration was initiated after the rats were stable on the rod. Each rat received 3 trials per testing day and the latency times to fall were recorded and averaged for each day.

### 4.5. Morris Water Maze Test

To evaluate the effect of PEG-bis-AA/HA-DXM treatment on cognitive function, we performed a Morris water maze (MWM) test. A circular pool (183 cm diameter) filled with water was maintained at 25 °C and located in a dimly lit room. Non-toxic black paint, Prang (Dixon Ticonderoga company, Lake Mary, FL, USA), was added to the water to make it dark and opaque. For evaluating the spatial learning and memory function, visual cues were placed external to the maze within the line of sight of the rats. A circular escape platform (15 cm diameter) was placed in a fixed location in the pool (southwest quadrant, SW) and submerged 2 cm below the water’s surface. Trials began at 8 DPI and were continued for 5 consecutive days (4 acquisition trials/day). For each trial, the rat was released from a randomized starting point (northwest (NW), north (N), northeast (NE), east (E), and southeast (SE)) facing the side of the pool. Rats were given a maximum of 60 sec to find the platform. If the rat found the platform, the rat would remain on the platform for 10 sec before being placed in its home cage. If the rat failed to find the platform within 60 sec, the rat was guided to the platform and allowed 10 sec to remain on the platform. During the trials, the swim paths were recorded using a video tracking system (SMART, Panlab, Harvard Apparatus, Cambridge, MA, USA), and the searching time to find the platform was recorded.

### 4.6. RT-PCR for Gene Expression

A real-time reverse transcription polymerase chain reaction (RT-PCR) was performed to examine the effect of PEG-bis-AA/HA-DXM on mRNA expression of inflammatory cytokines and apoptotic regulatory genes. At 14 DPI, the rats from each group were euthanized by CO_2_ overdose and the brains were retrieved. Ipsilateral cortex tissue (2 to 5 mm posterior of the bregma) was collected using a brain matrix (Electron Microscopy Sciences, Hatfield, PA, USA) and snap frozen in liquid nitrogen. Total RNA was extracted using a NucleoSpin RNA/Protein kit (Macherey-Nagel, Düren, Germany), according to the manufacturer’s method. The quality and quantity of isolated total RNA were measured using a Take 3 microplate reader (Biotek Instruments, Winooski, VT, USA). For the reverse transcription reaction, 1 µg of total RNA for each sample was used to synthesize cDNA using the qScript^TM^ cDNA Synthesis Kit (Quanta BioSciences Inc., Beverly, MA, USA), according to the manufacturer’s protocol. Real-time PCR was performed using a QuantiTect SYBR Green PCR Kit (Qiagen, Valencia, CA, USA) using custom-designed primers for the inflammatory cytokines *IL-1β* and *IL-10*, *TGF-β1*, *TNFα*, *IFN**γ* and apoptotic genes *Bcl-2* and *BAX* in a Rotor-Gene Q thermal cycler (Qiagen). Relative expression levels of target genes were quantified by the 2^−ΔΔCt^ method. Glyceraldehyde-3-phosphate dehydrogenase (GAPDH) was used as an endogenous control and expression was calculated as relative fold change compared to the expression level of the sham animal group. Preliminary PCR reactions were performed using a serially diluted template to confirm an amplification efficiency approximately equal to 1. At the end of all amplification reactions, melt curve analysis was performed to ensure the formation of a single product. Primer sequences are shown in Table 2.

### 4.7. Histological Analysis

At 14 DPI, the rats were deeply anesthetized (*n* = 5/group) with 5% isoflurane gas and sacrificed via cardiac perfusion with 0.9% saline followed by 4% paraformaldehyde (PFA). The brains were retrieved, post-fixed in 4% PFA for 3 days at 4 °C, and incubated in 10, 20, and then 30% sucrose for 24 h in each solution at 4 °C. Tissues were flash frozen to a cryostat mount using IceIT™ freeze spray (Fisher Scientific, Hampton, NH, USA) and 30 µm thick coronal sections prepared by cryostat (CM1950, Leica Biosystems, Buffalo Grove, IL, USA). Coronal sections were stored at −20 °C in cryoprotectant solution (30% sucrose, 1% polyvinylpyrrolidone, and 30% ethylene glycol in 1X PBS) until use.

### 4.8. Lesion Volume Measurement

To evaluate the effect of PEG-bis-AA/HA-DXM hydrogel treatment on cavity size, evenly spaced (0.25 mm spacing) coronal sections were selected from 2 to 4.5 mm posterior of the bregma level for each rat and stained for Nissl bodies (*n* = 10 sections/rat, *n* = 5 rats/group). Briefly, coronal sections were stained with 0.1% cresyl violet, dehydrated by ethanol gradient, cleared with xylene, and coverslipped with resinous mounting media (Azer Scientific, Morgantown, PA, USA). Images of sections were captured using an inverted bright-field microscope (Leica Microsystems, Wetzlar, Germany). Cavity areas were measured using ImageJ software (version: 1.53t, Wayne Rasband, National Institutes of Health, Bethesda, MD, USA) and the lesion volume was calculated by Cavalieri’s approximation according to the following equation:$$V=d\left(\overset{n}{\underset{i=1}{\sum }}y_{i}\right)-\left(t\right)y_{max}$$
where *d* represents the distance between measured sections, *y_i_* is the cross-sectional area of the lesion in the *i*-th section, *t* is the section thickness, and *y_max_* is the maximum possible value of *y*.

### 4.9. Immunohistochemical Staining for Neuroinflammatory Response, Neuronal Cell Survival, and Apoptosis

The effect of the PEG-bis-AA/HA-DXM hydrogel on the inflammatory response, apoptotic response, and neuron cell survival was assessed by immunohistochemistry. Briefly, sections were randomly selected from 3 to 4 mm posterior of the bregma (*n* = 3 sections/rat, *n* = 5 rats/group). Activated microglia/macrophages were detected using mouse monoclonal anti-macrophage/monocyte CD68 clone-ED1 antibodies (ED1, 1:200, MAB1435, EMD Millipore, Burlington, MA, USA), and neuronal nuclei were detected by mouse monoclonal anti-NeuN antibodies (1:200, MAB377, EMD Millipore, Burlington, MA, USA). Sections were incubated with the primary antibodies overnight at 4 °C prior to washing and incubation with Cy3-conjugated goat anti-mouse secondary antibodies (1:200, 115-165-003, Jackson Immunoresearch, West Grove, PA, USA). To detect apoptotic cells, terminal deoxynucleotidyl transferase-mediated dUTP nick-end labeling (TUNEL) staining was performed using an Apoptag^®^ Plus Fluorescein in situ Apoptosis Detection Kit (EMD Millipore, Burlington, MA, USA), according to the kit protocol. All slides were coverslipped with Vecta-Shield mounting media with DAPI, and images of the medial border of the lesion were captured using an Axiovert 40 CFL fluorescent microscope. ED1+, NeuN+, and TUNEL+ cells were counted along with the number of DAPI+ cell nuclei using ImageJ. To count DAPI+ cell nuclei, a threshold was applied to ImageJ to exclude the background signal, and a binary watershed was applied to separate touching nuclei. The number of nuclei was then counted by automatic particle analysis using ImageJ. ED1+, NeuN+, and TUNEL+ cells were manually counted by overlaying the DAPI signal to confirm that the staining signal corresponded to a cell. All events in each image were counted and the corresponding numbers of DAPI+ cell nuclei were used in calculating the percentage of ED1+ or TUNEL+ cells in each evaluated image.

### 4.10. Statistical Analysis

All data are presented as mean ± standard deviation (STD). Statistics for the lesion volume, ED1, and TUNEL analysis were conducted by Student’s *t*-test. Statistics for the rotarod, water maze, RT-PCR, and neuronal cell survival data were calculated using one-way ANOVA and pairwise comparisons were determined by Tukey’s HSD post hoc analysis. All statistical analyses were performed in GraphPad Prism 9.4.1 (San Diego, CA, USA). Adjusted *p*-values were calculated in the software and a *p*-value less than 0.05 was considered significant.

## 5. Conclusions

In this study, we demonstrated that the sustained local delivery of low-dose dexamethasone by PEG-bis-AA/HA-DXM hydrogels can improve motor function at 7 DPI by rotarod study, and this datum is consistent with our previously published evaluation of motor function by beam walk (ref). We also demonstrated that PEG-bis-AA/HA-DXM hydrogels can improve cognitive function at 14 DPI, as well as reduce neuroinflammation and apoptosis at 14 DPI in a rat mild-TBI model. In the future, we will evaluate the effects of PEG-bis-AA/HA-DXM hydrogels in a moderate or severe CCI TBI model as well as delayed treatment on secondary injury and motor and cognitive functions.

## Figures and Tables

**Figure 1 ijms-23-11153-f001:**
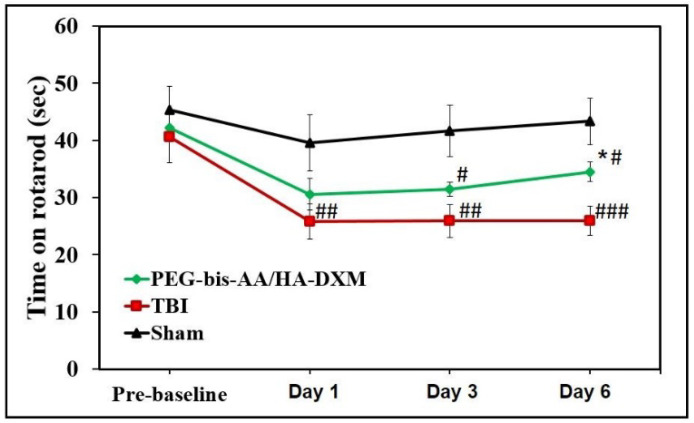
Effect of PEG-bis-AA/HA-DXM hydrogels on motor function by rotarod test. (*n* = 9 rats PEG-bis-AA/HA-DXM, *n* = 8 rats untreated TBI, *n* = 8 rats sham; data presented as mean ± STD; ONE-WAY ANOVA, post hoc: Tukey HSD; ^#^
*p* < 0.05, ^##^
*p* < 0.01, ^###^
*p* < 0.001 compared to sham; * *p* < 0.05 compared to TBI).

**Figure 2 ijms-23-11153-f002:**
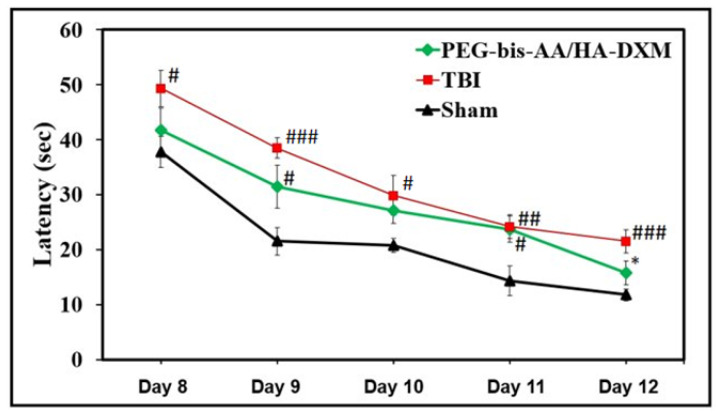
Effect of PEG-bis-AA/HA-DXM hydrogels on cognitive function by Morris water maze test with a hidden platform submerged in the southwest quadrant. Starting at 8 DPI, rats received 4 trials each day for 5 consecutive days. For each trial, each rat was allowed 60 s to search the pool for the submerged, hidden platform for all treatment groups; data presented as mean ± SD; ONE-WAY ANOVA, post hoc: Tukey HSD; ^#^
*p* < 0.05, ^##^
*p* < 0.01, ^###^
*p* < 0.001 compared to sham; * *p* < 0.05 compared to TBI).

**Figure 3 ijms-23-11153-f003:**
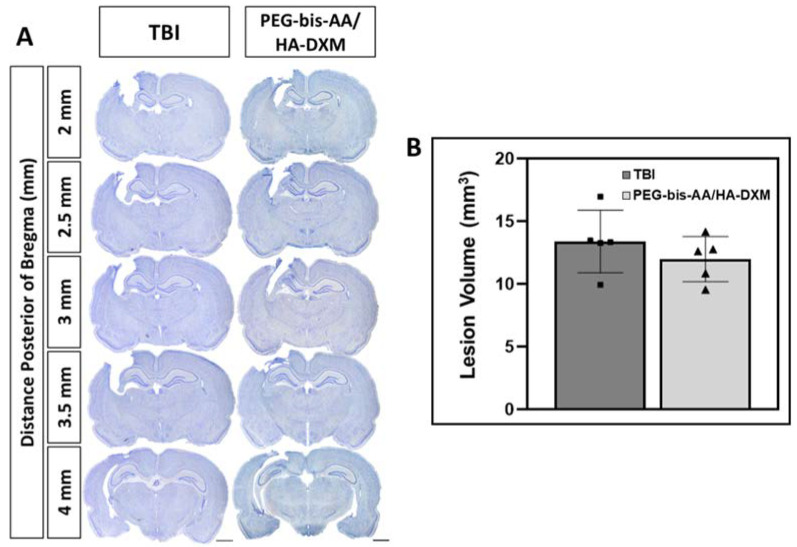
The effect of PEG-bis-AA/HA-DXM hydrogel treatment on average lesion volume at 14 DPI by Nissl staining. (**A**) Representative images of Nissl-stained sections are shown for 2 mm, 2.5 mm, 3 mm, 3.5 mm, and 4 mm posterior of the bregma. Scale bar = 2 mm. (**B**) Average lesion volume calculated by Cavalieri approximation. A total of 10 evenly spaced sections per animal (0.25 µm spacing, 2 mm to 4.25 mm posterior of the bregma). (*n* = 5 rats/group; data presented mean ± SD, ONE-WAY ANOVA, post hoc: Tukey HSD).

**Figure 4 ijms-23-11153-f004:**
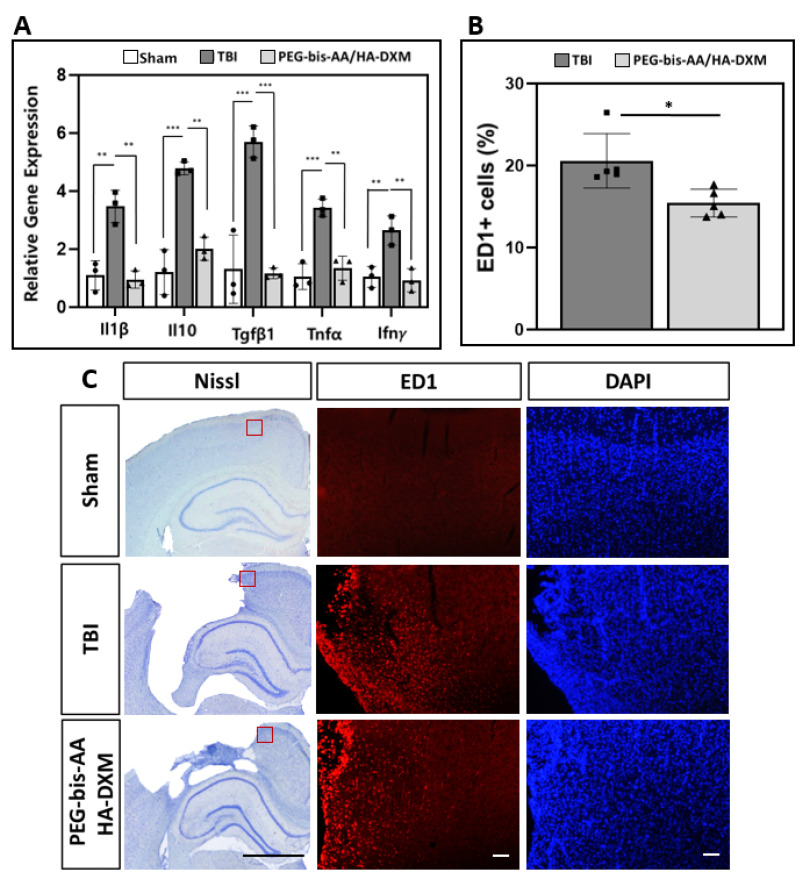
The effect of PEG-bis-AA/HA-DXM treatment on the neuroinflammatory response at 14 DPI. (**A**) The mRNA expression of the inflammatory cytokines *IL1β*, *IL10*, *TGFβ1*, *TNFα*, and *Ifn**γ*. *GAPDH* was used as an endogenous control, and expression levels were calculated by the 2^−ΔΔCt^ method (*n* = 3 rats/group). (**B**) The average percentage of ED1^+^-activated microglia/macrophages to total DAPI ^+^ nuclei (*n* = 3 sections/rat, *n* = 5 rats/group). Data are presented as mean ± STD. * *p* < 0.05, ** *p* < 0.01, *** *p* < 0.001 compared with TBI. (**C**) Representative images of ED1^+^ activated microglia/macrophages in the in ipsilateral cortex. Images are captured in the region indicated by the red box in the Nissl-stained brain section (black scale bar represents 2 mm). White scale bars represent 100 µm.

**Figure 5 ijms-23-11153-f005:**
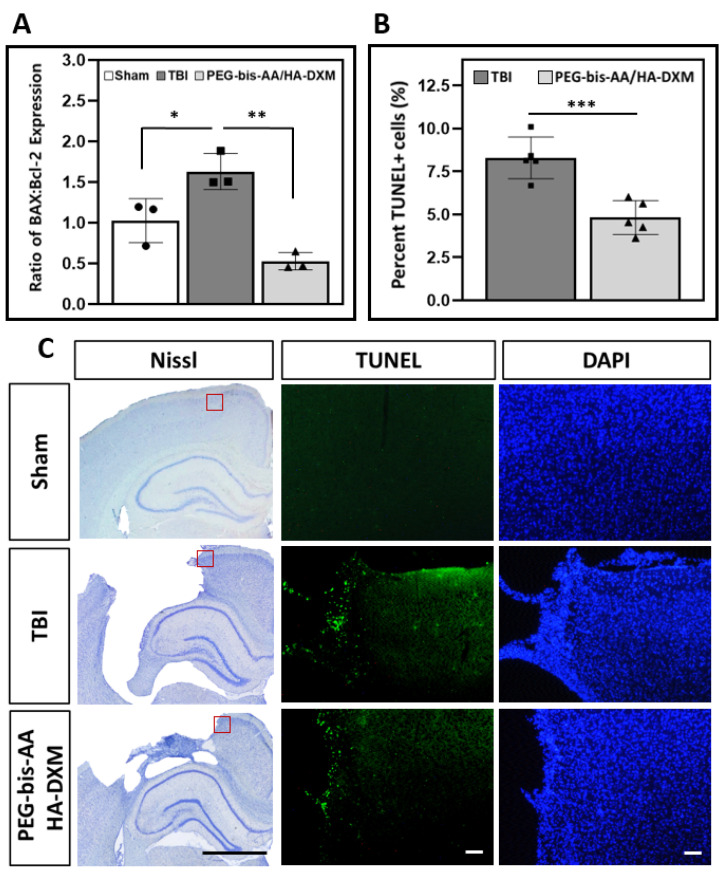
The effect of PEG-bis-AA/HA-DXM treatment on the apoptotic response at 14 DPI. (**A**) The ratio of the relative mRNA expression of BAX to Bcl-2 by RT-PCR (*n* = 3 rats/group). (**B**) The average percentage of TUNEL^+^ apoptotic nuclei (*n* = 15 sections/group, *n* = 3 sections/rat, *n* = 5 rats/group). Data are presented as mean ± STD. * *p* < 0.05, ** *p* < 0.01, *** *p* < 0.001 compared with TBI. (**C**) Representative images of TUNEL^+^ apoptotic nuclei in the ipsilateral cortex. Images are captured in the region indicated by the red box in the Nissl-stained brain sections (black scale bar represents 2 mm). White scale bars represent 100 µm.

**Figure 6 ijms-23-11153-f006:**
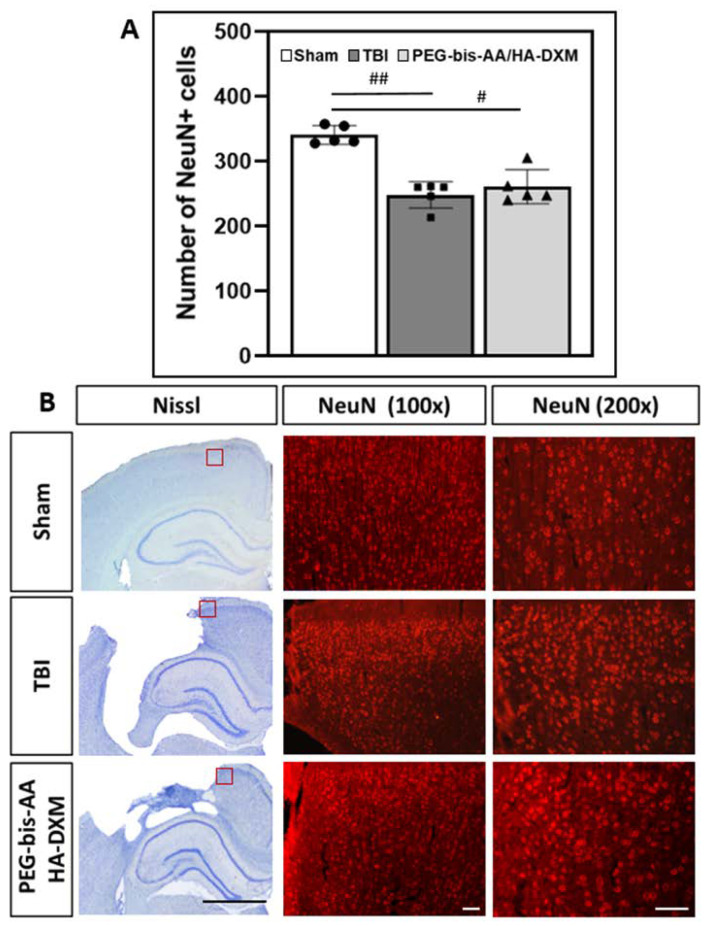
The effect of PEG-bis-AA/HA-DXM treatment on the apoptotic response at 14 DPI. (**A**) The average number of NeuN^+^ neuronal nuclei is counted in the image area (total *n* = 15 sections; *n* = 3 sections/rat, *n* = 5 rats/group). Data are presented as mean ± STD. ^#^
*p* < 0.05 and ^##^
*p* < 0.01 compared with sham. (**B**) Representative images of immunostaining for NeuN^+^ neuronal nuclei in the ipsilateral cortex. Images captured in region indicated by the red box in the Nissl-stained brain sections (black scale bar represents 2 mm). White scale bars represent 100 µm.

**Table 1 ijms-23-11153-t001:** Summary of key findings on the effect of PEG-bis-AA/HA-DXM hydrogel treatment from our previous study at 7 DPI [24] and the current study at 14 DPI.

Assessment	7 Days	14 Days
Lesion volume	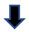	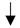
Macrophage/microglia infiltration	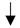	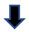
Reactive gliosis (GFAP)	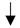	n/a
Apoptosis	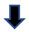	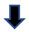
Inflammatory gene expression	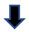	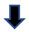
Neuronal cell preservation	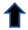	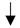
Motor function	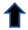	n/a
Cognitive Function	n/a	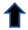

Note: Histological, molecular, and functional outcomes were evaluated for PEG-bis-AA/HA-DXM hydrogel treatment relative to untreated sham control at 7DPI [24] and 14 DPI. Changes in each outcome assessment within each individual study are indicated as statistically significant (thick arrows), qualitative or not statistically significant (thin arrows), or not assessed (n/a).

**Table 2 ijms-23-11153-t002:** Details for the primer sequences used in RT-PCR analysis of inflammation- and apoptosis-related genes.

Gene	Forward Primer (5′-3′)	Reverse Primer (5′-3′)	GeneBank No.	Product Size (bp)
IL1β	AGCTCTCCACCTCAATGGAC	GCCGTCTTTCATCACACAGG	NM_031512.2	148
TGFβ1	GTCAACTGTGGAGCAACACG	AGCCACTCAGGCGTATCAG	NM_021578.2	105
TNFα	AAGCATGATCCGAGATGTGG	AGGAATGAGAAGAGGTGAGG	NM_012675.3	105
IFN	TTGAAAGACAACCAGGCCATC	GCTTTGTGCTGGATCTGTGG	NM_138880.2	149
IL10	CAATAACTGCACCCACTTCCC	GTCAGCAGTATGTTGTCCAGC	NM_012854.2	120
BAX	GGATCGAGCAGAGAGGATGG	CAATTCGCCTGAGACACTCG	NM_017059.2	104
BCL2	GGAGGATTGTGGCCTTCTTTG	TCAGGTACTCAGTCATCCACAG	NM_016993.1	111
GAPDH	ATGGCCTTCCGTGTTCCTAC	AACTTTGGCATCGTGGAAGG	NM_107008.4	111

## Data Availability

The data presented in this study are available upon request from the corresponding author.

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
