# Peer review of "Dexamethasone-Loaded Hydrogels Improve Motor and Cognitive Functions in a Rat Mild Traumatic Brain Injury Model"

_ijms, 2022, doi:10.3390/ijms231911153_

Round 1

Reviewer 1 Report

The manuscript by Macks et al. describes the longer-term therapeutic efficacy of dexamethasone-loaded hydrogels on motor and cognitive functional recovery and secondary injury in rat mild controlled cortical impact (CCI) traumatic brain injury (TBI) model. The study is the extension of their previous study, where they assessed similar therapeutic efficacy but after 7 days post-injury (DPI). The study lacks novelty and whether providing dexamethasone-loaded hydrogels for a longer duration has better therapeutic efficacy. Based on this, I have the following concerns:

Major comments:

1.    It is not clear from the introduction how different and important is the current study. A well-defined rationale is required.

2.    A comparison between day 7 and day 14 is required to highlight how different the therapeutic efficacy is between the parameters studied.

Author Response

1. It is not clear from the introduction how different and important is the current study. A well-defined rationale is required.

Res) Thank you for constructive comment. We have revised the last paragraph of the introduction to clarify the rationale and importance of the study, particularly with respect to our previous work.

2. A comparison between day 7 and day 14 is required to highlight how different the therapeutic efficacy is between the parameters studied.

Res) Thank you for emphasizing the importance of comparison of therapeutic efficacy between 7 and 14 DPI. We have discussed our observation in discussion section -Please see the highlights.

Reviewer 2 Report

The study team provides an interesting research and I do think it will help in the treatment of TBI. I still have some concerns and here are my comments.

1.        Please provide the raw gel of the RT-PCR experiments.

2.        Please replace the bar plots with dot plots, which would be helpful to exhibit the data distribution.

3.        Figure 4. As for the ED1+ cell percentage, the study team could perform the analysis based on double-labeling of ED1 with Iba1 instead of DAPI, because CD68+ reflects the function of microglia.

4.        Figure 5. The same as figure 4. It would be nice to perform the analysis on the immunostaining of TUNEL and Neun, which could better demonstrate the apoptosis of neurons. The death of neurons is directly related to the cognitive deficit.

5.        Figure 5 and figure 6. A bar is also needed for images with nissl and DAPI staining.

6.        Figure 6. The background of the images is high. Please replace them with the ones with lower background.

7.        The authors did not describe the method for the cell counting. What kind of software did they use? What’s the strategy? Please state the information in detail.

Author Response

The study team provides an interesting research and I do think it will help in the treatment of TBI. I still have some concerns and here are my comments.

1. Please provide the raw gel of the RT-PCR experiments.

Res) We did not perform electrophoresis of the PCR products. Amplification and quantification was performed by real-time PCR using SYBR green quantification. Melting curve analysis was performed at the end of each group of reactions to verify as single peak in dF/dT, demonstrating formation of a single product.

2. Please replace the bar plots with dot plots, which would be helpful to exhibit the data distribution.

Res) Thank you for the suggestion. Dot plots have been overlaid on the bar plots to provide additional insight into data distribution as suggested.

3. Figure 4. As for the ED1+ cell percentage, the study team could perform the analysis based on double-labeling of ED1 with Iba1 instead of DAPI, because CD68+ reflects the function of microglia.

Res) We appreciate the reviewer’s constructive comment, however, for this current study, the number of remaining sections from the region of interest are not sufficient for performing the requested analysis. The authors will include double-labeling of ED1 and Iba1 for characterizing microglia activity in the future studies.

4. Figure 5. The same as figure 4. It would be nice to perform the analysis on the immunostaining of TUNEL and Neun, which could better demonstrate the apoptosis of neurons. The death of neurons is directly related to the cognitive deficit.

       Res) The reviewer raises a very good point. Although we show that cortical cells are undergoing apoptosis (TUNEL) and that there is neuronal cell loss (NeuN), our current analysis does not allow insight into the causal role of apoptosis in neuronal cells loss. We appreciate the reviewer’s suggestion and will include this type of analysis in future studies.

5. Figure 5 and figure 6. A bar is also needed for images with nissl and DAPI staining.

Res) Figure 5 has been updated including a scale bar in the DAPI staining image. A scale bar for the nissl images in Figure 4-6 has been added.

6. Figure 6. The background of the images is high. Please replace them with the ones with lower background.

Res) The author’s have re-checked the images used in analysis for Figure 6 to find an image with lower background, however, all sections exhibit relatively similar background level in NeuN staining. During our staining found that the background is typically higher at the edge of the lesion cavity when staining for NeuN. We believe that this could be due in part to the primary antibody considering the secondary antibody used for NeuN staining is the same as for ED1 staining. We are hesitant to adjust brightness/contrast in order to attempt to reduce the background near the lesion boundary as this might excessively diminish staining intensity in other regions of the sections. Therefore, we think that presenting the staining as originally imaged without adjustment is best.

7. The authors did not describe the method for the cell counting. What kind of software did they use? What’s the strategy? Please state the information in detail.

Res) In consideration of the reviewer’s question, we have updated the text starting at line 462 to include a more detailed description of the method used for counting ED1, TUNEL, NeuN, and DAPI events in staining images. However, the authors would like to mention that a short description of the method used for cell counting was included in the original text starting at line 461.

Reviewer 3 Report

1. Materials and methods, 4.3: Surgical manipulation of the dura mater should be described; in CCT the dura mater is usually spared and external force is applied over the dura mater. Describe the surgical manipulation of the dura in all experimental, TBI, and sham groups.

2. Materials and methods, 4.3: The TBI group would have required placing drug-free hydrogengel on the brain surface. A description of why it was not placed is required.

3. Abstract: The abstract ends with a description of the results. The main points of the conclusion should be stated at the end of the abstract.

Author Response

1. Materials and methods, 4.3: Surgical manipulation of the dura mater should be described; in CCT the dura mater is usually spared and external force is applied over the dura mater. Describe the surgical manipulation of the dura in all experimental, TBI, and sham groups.

Res) In the Materials and Methods Section 4.3, we described the manipulation of dura in sham, TBI untreated, and hydrogel treatment groups with track changes and highlights.

2. Materials and methods, 4.3: The TBI group would have required placing drug-free hydrogel on the brain surface. A description of why it was not placed is required.

Res) Thank you for your comments. We addressed this point in discussion section with the following statement, “We did not include a PEG-bis-AA/HA hydrogel without DX group because we did not observe significant therapeutic benefit in our previous study at 7 DPI [24].”. For the better understanding/justification, we added this sentence to “Materials and Methods” section as well.

In our previous published paper, we placed PEG-bis-AA/HA hydrogel containing native HA (no dexamethasone) as a control and we observed that this drug-free hydrogel showed slightly improved motor function, reduced lesion size, reduced apoptosis, and improved neuronal survival compared to untreated TBI, but without significant difference. Therefore, we did not include drug-free hydrogel in this study.

3. Abstract: The abstract ends with a description of the results. The main points of the conclusion should be stated at the end of the abstract.

Res) Thank you for the constructive comment. We added the conclusion sentence, “Therefore, PEG-bis-AA/HA-DXM hydrogels can be promising therapeutic intervention for TBI treatment.” at the end of abstract.

Round 2

Reviewer 1 Report

The authors have tried to address all my raised concerns, however, I believe adding a comparative table between the previous findings and current results will enhance the current manuscript.

Author Response

The authors have tried to address all my raised concerns, however, I believe adding a comparative table between the previous findings and current results will enhance the current manuscript.

Res) Thank you for your constructive comment. We added a table as reviewer suggested and wrote “Table I summarizes our key findings on the effect of PEG-bis-AA/HA-DXM hydrogel treatment from our previous study at 7 DPI and the current study at 14 DPI. Overall, these results show a consistent and sustained therapeutic effect.” in the discussion.

Reviewer 2 Report

It can be accepted now.

Author Response

We appreciate your acceptance of our manuscript.